# Integration of Multiple Resolution Data in 3D Chromatin Reconstruction Using *ChromStruct*

**DOI:** 10.3390/biology10040338

**Published:** 2021-04-16

**Authors:** Claudia Caudai, Monica Zoppè, Anna Tonazzini, Ivan Merelli, Emanuele Salerno

**Affiliations:** 1National Research Council of Italy, Institute of Information Science and Technologies, 56124 Pisa, Italy; anna.tonazzini@isti.cnr.it (A.T.); emanuele.salerno@isti.cnr.it (E.S.); 2National Research Council of Italy, Institute of BioPhysics, 20133 Milano, Italy; monica.zoppe@cnr.it; 3National Research Council of Italy, Institute of Biomedical Technologies, 20054 Segrate, Italy; ivan.merelli@itb.cnr.it

**Keywords:** chromatin conformation, bayesian statistics, HI-C data, chromatin conformation capture, CTCF CHIA-PET data, CHIP-seq, RNA-seq

## Abstract

The three-dimensional structure of chromatin in the cellular nucleus carries important information that is connected to physiological and pathological correlates and dysfunctional cell behaviour. As direct observation is not feasible at present, on one side, several experimental techniques have been developed to provide information on the spatial organization of the DNA in the cell; on the other side, several computational methods have been developed to elaborate experimental data and infer 3D chromatin conformations. The most relevant experimental methods are Chromosome Conformation Capture and its derivatives, chromatin immunoprecipitation and sequencing techniques (CHIP-seq), RNA-seq, fluorescence in situ hybridization (FISH) and other genetic and biochemical techniques. All of them provide important and complementary information that relate to the three-dimensional organization of chromatin. However, these techniques employ very different experimental protocols and provide information that is not easily integrated, due to different contexts and different resolutions. Here, we present an open-source tool, which is an expansion of the previously reported code *ChromStruct*, for inferring the 3D structure of chromatin that, by exploiting a multilevel approach, allows an easy integration of information derived from different experimental protocols and referred to different resolution levels of the structure, from a few kilobases up to Megabases. Our results show that the introduction of chromatin modelling features related to CTCF CHIA-PET data, histone modification CHIP-seq, and RNA-seq data produce appreciable improvements in *ChromStruct*’s 3D reconstructions, compared to the use of HI-C data alone, at a local level and at a very high resolution.

## 1. Introduction

The three-dimensional organization of chromatin is involved in regulation of gene function and connected with physiological and pathological correlates and dysfunctional cell behavior. A advancement in chromatin studies was made possible with the development of next-generation sequencing techniques [1], which enabled a number of methods of Chromosome Conformation Capture and its derivatives, such as HI-C techniques [2]. These techniques provide information on how frequently the possible chromatin fragment pairs are in close contact in a population of cells. Depending on the experimental details and on the restriction enzyme used, the genomic resolution of HI-C can range from a few kilobases to Megabases. The opportunity to observe the chromatin structure at multiple resolutions emerged recently, following investigations on the 3D structure of TADs (Topologically Associating Domains) [3], broadly defined as portions of chromatin with more internal than external interactions, are sometimes described as fairly isolated *globular* structures. TADs have a dynamic nature and play a role in gene expression and maintenance of cellular identity [4]. Chromatin conformation is also associated with metabolic activity: for example, transcription requires the DNA to be accessible to a large number of enzymes, involved in all steps: from regulation to intiation, and to progression of the RNA polymerases, often in multiple copies, along the transcribed DNA portion, extending up to many tens of kilobases. This activity, in which the epigenetic code of histone modifications has a role, as reviewed in [5], implies different degrees of chromatin compaction.

In addition to HI-C, other experimental techniques such as CHIP-seq using specific antibodies and RNA-seq can provide information on geometrical features of chromatin [6,7,8,9,10]. CHIP-seq experiments allow the characterization of genomic loci by their association with specific proteins, such as transcription factors or other DNA binding proteins (e.g., CTCF), or by finer molecular details, such as histonic modifications, (e.g., acetylation, mono- or three-methylation at specific histone sites), all of which are associated with distinct functional genomic features. RNA-seq experiments provide information on which DNA loci have been transcribed, identifying genomic portions that are accessible to the transcription machinery, and loose enough to allow exposure of the sequence. Careful consideration of all the information available suggests that some data can (or should) be redundant, while others are complementary or mutually explanatory. In [11], Lieberman Aiden et al. showed, by Principal Component Analysis, that the distribution of contacts in the HI-C matrices correlates with the distribution of genes and with features of open or silent chromatin. For example, expressed genes are frequently marked by a chromatin state that includes H3K27AC modification in their enhancer region and H3K4ME3 in their promoter region, while repressed genes are often marked by the H3K27ME3 modification in their body [12,13]. However, not all expressed genes bear such a mark, and not all marked genes are necessarily expressed.

The method proposed here enables the elaboration of plausible 3D chromatin conformations through the integration of several pieces of information. The integration of data at different resolutions permits the derivation of structural properties that are not easily deduced using the different data separately. A number of computational methods have been developed to determine the 3D structures of chromosomes from contact-frequency matrices [14,15,16,17]. Many of these methods transform contact frequencies into Euclidean distances, and then reconstruct the structure by solving a distance-to-geometry problem (for example, see [17]). This frequency-to-distance transformation, however, presents a major problem, as it invariably produces geometrically inconsistent distance sets [18,19]. To avoid the drawbacks of this strategy, we reject the derivation of distances from the contact frequencies, and adopt an iterative multiscale procedure to derive the 3D structure directly from the contact frequencies. In previous work [20], we introduced *ChromStruct*, a method to infer a set of spatial chromatin conformations starting from the contact information of HI-C experiments. This method is based on a multiscale chromatin chain model made of consecutive and partially penetrable beads of different sizes. The algorithm automatically divides the contact matrix into variable-size diagonal blocks, and reconstructs the related 3D structures independently for each block. This is made possible by the fact that the chromatin chain presents regions, such as the TADs, with many internal interactions between pairs of loci and interact much less with other regions of the chain. Each diagonal block is used to estimate the structure of the related sub-chain by sampling a solution space generated by a score function based on both data-fit and implicit, soft, geometrical constraints. The sub-chains thus obtained are then modeled as single beads in a coarser-scale chain and the procedure iteratively repeats following the same rules used for the finer scales, until no more isolated blocks are detected in the binned data matrix, i.e., the entire chromosome is modeled. The whole chain is then reconstructed iteratively from the coarsest to the finest scale, by substituting each bead with the corresponding sub-chain at the finer scale, maintaining its 3D orientation (see Figure 1). The intention of our sampling of the solution space is not to find a unique consensus, but a family of solutions. This is consistent with the fact that *hi-c* data derive from millions of cells, where different configurations contribute. The *ChromStruct* sampling strategy does not search for a global minimum, but explores the solution space to find a number of configurations with similar scores. An advantage of the *ChromStruct* strategy is that the score function is designed to allow the user to introduce and integrate different features and data sources, even at different resolution levels.

In this paper, we present an extension of *ChromStruct*, which allows the HI-C data to be integrated with data derived from other experimental techniques, and demonstrate its use with histone modification specific CHIP-seq, RNA-seq and CTCF CHIA-PET data. The algorithm browses different resolution levels, from the smallest sub-TAD to the entire chromosome, enabling the investigation of the details of folding inside the TADs, at the intermediate structures of nested domains, and at the macroscopic organization of the compartments at the coarsest scale. HI-C experiments provide information on contact frequencies between portions of the chromatin fiber; histone mark CHIP-seq data provide additional information about DNA geometry and 3D occupancy; RNA-seq data provide information about gene expression and, therefore, on the compactness of DNA; finally, CTCF CHIA-PET data inform us about loops that bring distant genome elements into spatial proximity. We introduce these data in the score function as geometrical information. Our results show that the introduction of CTCF CHIA-PET data, RNA-seq and CHIP-seq information can produce more detailed conformations at very high resolutions (few kb), while at lower resolution, such improvement is not perceived.

## 2. Results

In this work, we describe an extension of *ChromStruct* to integrate information derived from HI-C experiments with further geometrical data derived from histone specific modification (H3K27ME3) CHIP-seq, RNA-seq and CTCF CHIA-PET experiments. These data are obtained through different laboratory approaches; this is advantageous from a scientific point of view, because they refer to conformational information from different perspectives. For example, portions of DNA showing a high degree of H3K27 three-methylation are enriched in repressed genes and are, therefore, more compact [13], whereas portions where RNA-seq signal the presence of expressed genes are more expanded. A score function that can be interpreted as a log-posterior probability evaluated by applying the Bayes rule manages the relevance of the geometric information available, rewarding the configurations that are consistent with the data of different nature and penalizing the ones that appear discordant or uncertain as geometrical interpretations of multiple experimental data. *ChromStruct* samples the solution space generated by this score function by an approximated simulated annealing [21]. In our experiments, we considered chromosome 12 of human hematopoietic progenitor cells. We collected HI-C contact matrix at a 5 kb resolution, RNA-seq data, H3K27ME3 CHIP-seq data at 20 bp resolution [8] and CTCF CHIA-PET data [22] (see Section 4 for details). With these data, we compared the *ChromStruct*’s reconstructions using HI-C data alone and using HI-C data integrated with H3K27ME3-CHIP-seq, RNA-seq and CTCF CHIA-PET data.

### 2.1. High-Resolution Configurations

*ChromStruct* takes HI-C contact frequency matrices as the first input. These can be very large (as in the case of chromosome 12 at 5 kb resolution), and in order to manage the amount of data and lower the computational costs, the first step of the algorithm consists in the division into blocks, respecting the fractal structure of the TADs [21]. The block-detection algorithm (based on Moving Average on sliding triangles on the main diagonal, described in detail in [21,23]) found 2097 diagonal blocks with an average genomic size of 12 fragments of 5 kb (i.e., 60 kb). To analyze the behavior of the new score-function at a 5 kb resolution, we selected a smaller 3.5 Mb portion of chromosome 12, containing 50 blocks, and a variety of chromosomal features, as reported in Table 1. As shown, 16 blocks are interested by Histone 3 Lysine 27 three-methylation, 9 by gene expression and in 7 blocks, we have CTCF-mediated internal contacts. Because repression and expression have opposite effects, the score-function annihilates their contribution in the few blocks in which both are contained.

The score function allows the fiber’s curvature to be higher in portions of chromatin with H3K27ME3 and penalizes high curvatures in portions interested by expressed genes. The bead diameters also depend on whether they belong to expressed or silent areas (see Section 4 for details). The CTCF feature is modeled as an increase in contact frequency within the HI-C matrix, for the pairs characterised by CTCF-coupling, so as to represent a stronger constraint for their proximity (see Section 4 and Appendix A). Figure 2 shows the comparison between the distributions of correlations between the contact matrices calculated from the estimated configurations and the original HI-C contact matrix for the block in the portion of chromosome 12 considered. In the left panel, we consider the blocks interested by active genes, in the central panel the blocks interested by H3K27ME3, associated with repression and, in the right panel, all blocks with CTCF mediated contacts. The boxplots show that, at a 5 kb resolution, the integration of geometrical information other than HI-C contacts improves the correlation between the synthetic contact matrices and the original contact matrix. The introduction of additional geometrical constraints in our experiments thus allows, at this resolution level, the reconstruction of more accurate high-resolution configurations.

### 2.2. Low-Resolution Configurations

Our algorithm reconstructs the chains at lower resolutions by binning the blocks at the current resolution level and associating them into single beads in the lower-resolution chain [24] with sizes derived from the three-dimensional structures at the previous level. As shown in Table 2, the fine details at a 5 kb resolution are not visible at lower resolutions. Indeed, the information related to CHIP-seq and RNA-seq is only detected at the same resolution for which these data are relevant. At lower levels, their introduction no longer highlights significant differences compared to the use of HI-C data only. The correlation between the original HI-C contact matrix of the whole chromosome 12 and the synthetic contact matrices obtained by pooling 100 conformations generated by *ChromStruct* using HI-C data only and 100 conformations generated using HI-C, H3K27ME3-CHIP-seq, RNA-seq and CTCF CHIA-PET data, show no significant differences. A reconstruction of the selected portion of chromosome 12 [111.5 Mb–115 Mb] at a 5 kb resolution is represented in Figure 3a; a reconstruction of the whole chromosome at 500 kb resolution is shown in Figure 3b.

Two main reasons can explain the different behaviour at high- and low-resolution. First, the information contained in the HI-C contact matrix and the one derived from histone-mark immuno-precipitation sequencing and RNA-sequencing are not independent. This can be seen in Figure 3c, where the right-hand side, with higher contact density (more yellow in the contact matrix), contains fewer genes (ENCODE tract), and is more interested by Histone 3K27 three-methylation. The opposite is visible in the part on the left. As Lieberman-Aiden et al. demonstrated in [11], HI-C contact matrices already contain a lot of structural information correlated with the distribution of genes and with the features of open chromatin. The second reason is that *ChromStruct*, in its multi-level approach, already takes into account the existing correlation between contact density and compactness of the chromatin fiber. Specifically, *ChromStruct* sets the size of every bead in inverse proportion to its number of contacts [21]. Blocks with many contacts reasonably correspond to more compact fiber portions, while blocks with few contacts are likely to correspond to more expanded and more easily transcribed areas. As in large-scale geographical maps, the fine details, such as minor roads or buildings, are not visible but appear when the scale is progressively refined. In our case, the presence of high-resolution details does not affect the overall appearance at large scales, which is completely determined by low-resolution data.

From our experiments, we observe that *ChromStruct*, equipped with the score function described in Section 4, makes it possible to investigate the spatial organization to a high degree of detail, introducing more precise information in the reconstruction at very high resolution (5 kb). Moving to lower resolutions, these details are not delineated; however, the macroscopic structure and the various dimensions remain consistent with the structural information derived by HI-C contact matrices alone.

## 3. Discussion

The organization of chromatin at the resolution levels between the wrapping of DNA around histones and the chromosomal domains is not yet completely explained. Experiments of Chromosome Conformation Capture, and in particular those of HI-C type, have contributed to consistent hypotheses on the organization of chromatin within the nucleus. The relationship between the chromatin 3D structure and epigenetic states has been highlighted since the early times of *Chromatin 3D* studies [3] and has been exploited to confirm the validity of 3D reconstructions and to derive further information on chromatin biological features [25,26,27,28]. However, the introduction of epigenetic information *a priori*, as a means for the more detailed elaboration of 3D reconstruction, has been attempted only recently [12,29,30,31]. As many of the experiments on epigenetic features provide information at different levels of resolution, a multilevel approach appears necessary [15,32]. The introduction of different information at different resolution levels also tests the correctness of the available data; the Bayesian approach reinforces information that is consistent from a topological point of view and tends to cancel information coming from conflicting inputs. From our experiments, it emerged that the introduction of data related to three-methylation of Histone 3 Lysine 27, gene expression and CTCF-mediated coupling in the score-function of *ChromStruct* allows the reconstruction of more accurate conformations at a local level, at a resolution of the same order of magnitude as the data introduced.

## 4. Materials and Methods

### 4.1. Data Origin and Treatment

The HI-C, CHIP-seq and RNA-seq data used for the experiment refer to human CD34 hematopoietic progenitor cells (GM12878) [8,33]. Data on CTCF-mediated coupling, obtained through CHIA-PET experiments, were downloaded from GEO accession number GSM1872886. To create contact frequency matrices, *fastq* data were translated into *sam* format with the BOWTIE2 program, using the HG19 alignment as reference genome. Through HICEXPLORER, the *sam* files were first transformed into *bam* format, and then into *Hierarchical Data Format*. We used the R package DIFFHIC [34] (choosing Bonferroni correction to lower the False Discovery Rate [35]) to create contact matrices at 5 kb.

In order to introduce information derived from Chromatin IP and RNA-seq, it is necessary to consider the size resolution of these data. Data from RNA sequencing experiments provide information on transcribed genes; these were introduced as a binary 1-dimensional array at a 5 kb resolution (“1” if the bin is interested by expressed genes, “0” otherwise). In the case of the Histone 3 Lysine 27 three-methylation mark, which is associated with repressed genes [36], CHIP-seq data are reported at a resolution of 20 base-pairs, making it necessary to bin the data at the resolution of HI-C contact matrices (5 kb). Once binned, they are introduced as a binary 1-dimension array (“1” if the binned data point exceeds the threshold of 300, “0” otherwise). Finally, CTCF specific capture HI-C precipitation data inform about two DNA segments that are found in close contact. The segments can be at any genomic distance, and can span between TADs; therefore, this information can either be inserted at a 5 kb resolution, when the two DNA segments lie in the same TAD, or at a lower resolution, when the DNA segments involved belong to different TADs. CTCF CHIA-PET contact features are introduced as a binary matrix, at the same dimension and same resolution of the HI-C contact matrix (“1” if the couple of beads correspond to a binding site, “0” otherwise). This matrix, multiplied by a scalar factor and added to the HI-C contact matrix, enforces the proximity constraints for binding sites (further details in Appendix A). Features are modelled as binary 1- and 2-dimensional arrays in order to be easily introduced into the score-function; however, other modelling approaches are possible, perhaps weighing features distributions.

### 4.2. Volume Considerations

The interphase nucleus of a typical cell is a roundish structure of about 5 micrometers in diameter. The fraction of the volume actually occupied by chromatin (DNA + histone octamers) is about one third. Historical histological observations describe Eu- and Hetero-chromatin as two distinct states; recent CRYOEM studies [37] have further characterized this distinction, and shown a Chromatin Volume Concentration (CVC) that ranges from 12% to 50%. From these values, and following the distribution reported, we can attribute a CVC of 20% to euchromatin (the transcriptionally active portion of DNA) and of 40% to heterochromatin (the silent, repressed portion). Based on these considerations, we reduced the bead diameters by 10% in repressed portions and increased them by 10% in active portions. Our score-function is also designed to modulate the density of CVC to plus or minus 50%, respectively, for regions that contain expressed genes and silent regions, by decreasing and increasing the admissible curvature of the thread.

### 4.3. Solution Space Sampling

The score function that generates the solution space for each sub-chain C to be estimated has the following form:(1)$$\Xi \left(C\right)=\Phi_{HiC}\left(C\right)+\mu_{1}\Phi_{ChIP}\left(C\right)+\mu_{2}\Phi_{RNA}\left(C\right)+\lambda \Psi \left(C\right)$$
where $\Phi_{HiC}$, $\Phi_{ChIP}$ and $\Phi_{RNA}$ are the data-fit terms corresponding to HI-C contact data, CHIP-seq data and RNA-seq data, respectively, Ψ instead is the constraint term. Parameters $\mu_{1}$, $\mu_{2}$ and λ are intended to balance the mutual influence of the different terms.

The term $\Phi_{HiC}$ forces bead pairs with many mutual contacts to be close to each other:(2)$$\Phi_{HiC}\left(C\right)=\sum_{i,j\in L}n_{ij}{\left[d_{ij}-\left(r_{i}+r_{j}\right)\right]}^{2}$$
where $n_{ij}$ is the contact frequency of the *i*-th and *j*-th beads, $d_{ij}$ is the distance between their centroids, and $r_{i}$ and $r_{j}$ are their radii. L is a subset of contacts in each sub-chain, made of the pairs exceeding a pre-defined percentile of the contact frequencies in the related block.

By controlling the maximum distance between the beads in the sub-chain, the term $\Phi_{ChIP}$ increases the curvature of a sub-chain affected by methylation. Due to this term, regions with high concentration of Histone modification H3K27ME3 (associated to repression) are steered to be more compact:(3)$$\Phi_{ChIP}\left(C\right)={\left[\max_{i,j\in L_{\mathrm{CHIP}}}\left(d_{ij}\right)-d_{min}\right]}^{2}$$
where $L_{\mathrm{CHIP}}$ is the set of all pairs (with the exception of the pairs located on the two main diagonals) if the block is interested by H3K27ME3, the empty set otherwise. The value $d_{min}$ is derived as follows:(4)$$d_{min}\left(C\right)=d_{c}\frac{\sqrt[{5/6}]{RIS}}{6\pi }$$
and represents the estimate of the minimum size that a bead can assume at the resolution of RIS (in our case 5 kb) with the diameter of the chromatin fiber equal to $d_{c}$ (in our case 30 nm). The term $\Phi_{ChIP}$ gives little penalty if the maximum distance between the beads of a sub-chain is close to $d_{min}$.

The term $\Phi_{RNA}$ acts by controlling the minimum distance between non-adjacent beads in the subchain, thus reducing its curvature and makes regions with active genes more expanded:(5)$$\Phi_{RNA}\left(C\right)={\left[\min_{i,j\in L_{\mathrm{RNA}}}\left(d_{ij}\right)-d_{max}\right]}^{2}$$
where $L_{\mathrm{RNA}}$ is the set of all pairs if the block is interested by expressed genes, and the empty set if not. The value $d_{max}$ represents the estimate of the maximum size for a bead at the resolution of RIS and with a chromatin’s diameter of $d_{c}$:(6)$$d_{max}\left(C\right)=d_{c}\frac{\sqrt[{5/6}]{RIS}}{3\pi }$$

The term $\Phi_{RNA}$ gives little penalty if the minimum distance between the beads of a sub-chain (except for consecutive ones) is at least $d_{max}$.

When any two beads in C interpenetrate, one of the terms in brackets becomes negative. The maximum data-fit penalisation of this situation occurs when $d_{ij}=0$, and in Equation (Equation 2) $\varphi_{ij}$ assumes the finite and unmodifiable value $n_{ij}{\left(r_{i}+r_{j}\right)}^{2}$. The constraint term Ψ is needed to control this penalisation:$$\Psi \left(C\right)=\sum_{i,j\in C}\frac{r_{i}+r_{j}}{2d_{ij}}\left[1-\frac{{\left\{c\left[d_{ij}-\left(r_{i}+r_{j}\right)\right]\right\}}^{b}}{1+\left\{c\right|d_{ij}-\left(r_{i}+r_{j}\right){\left|\right\}}^{b}}\right]$$
where *c* is a scale factor that makes the terms in braces dimensionless, and the exponent *b* is an odd natural. For $d_{ij}$ near zero, $\psi_{ij}$ behaves as $\left(r_{i}+r_{j}\right)/d_{ij}$, whereas in an interval around $\left(r_{i}+r_{j}\right)$ it behaves as $\left(r_{i}+r_{j}\right)/\left(2d_{ij}\right)$ and, for $d_{ij}$ sufficiently larger than $\left(r_{i}+r_{j}\right)$, it goes rapidly to zero. Parameter *b* tunes the slope of the transitions between the different zones; large values of *b* produce abrupt transitions. Term Ψ is intended to prevent any two beads from interpenetrating more than some fraction of their sizes. Note that, when adjacent or genomically close beads are involved, modulating the allowed mutual interpenetration is also effective to avoid knots and to constrain the local curvature of the chain.

In order to allow easy setting of the geometric parameters related to the data resolution, the size of the beads, the diameter of the fibre, the mechanism of subdivision into TADs and the coefficients of the score function, *ChromStruct* is equipped with a Graphical User Interface (GUI). The GUI, illustrated in Figure 4, also allows users to insert the input data from dialog boxes: the HI-C matrix (required) and the CHIP-seq, RNA-seq and CTCF-binding arrays (optional). Guidelines for GUI’s use are detailed in the Appendix A.

## 5. Conclusions

Using multilevel approaches in computational biology is, in some cases, necessary: on the one hand, this strategy makes it possible to parallelise the algorithms, greatly reducing the computational cost. On the other hand, it becomes possible to introduce data obtained from experiments that produce results at different scales, and to analyse biological structures navigating between different scales.

We have made *ChromStruct* suitable for introducing information at different resolutions, so as to be able to exploit and integrate as much knowledge as possible on the three-dimensional organization of chromatin, also deriving from different biological experiments and also belonging to different dimensional scales.

## Figures and Tables

**Figure 1 biology-10-00338-f001:**
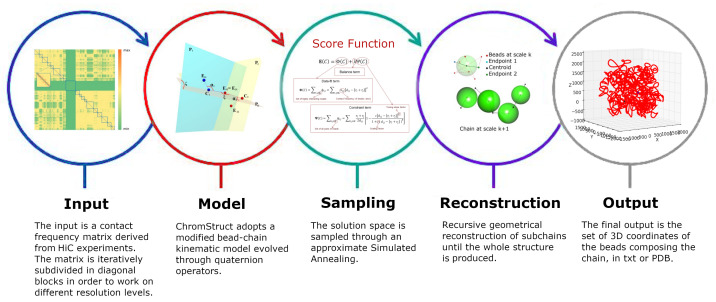
Flow of *ChromStruct*: (**blue**) the input HI-C contact frequency matrix is subdivided in diagonal blocks. (**red**) Chromatin fibre is modeled as a chain of partially penetrable beads and subdivided into sub-chains. (**green**) Geometrical perturbations are performed in the quaternion algebra and the solution space is sampled by a Bayesian method. (**violet**) As the last step, a multilevel 3D reconstruction generates chromatin output conformations (**gray**) that are compatible with input and constraints.

**Figure 2 biology-10-00338-f002:**
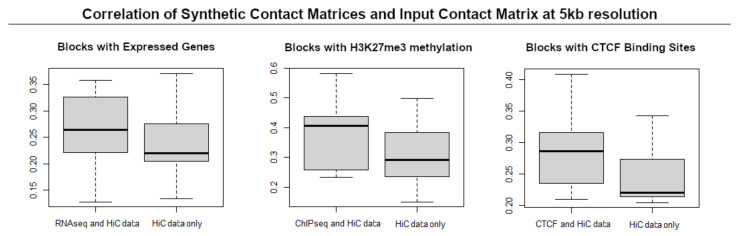
Comparison of distributions of Pearson correlation between contact matrices obtained with *ChromStruct* and original contact matrix for blocks belonging to a 3.5 Mb portion of chromosome 12. Blocks interested by expressed genes (**left**), H3K27ME3 (**centre**), and CTCF CHIA-PET (**right**) show a higher correlation if the relevant information is used.

**Figure 3 biology-10-00338-f003:**
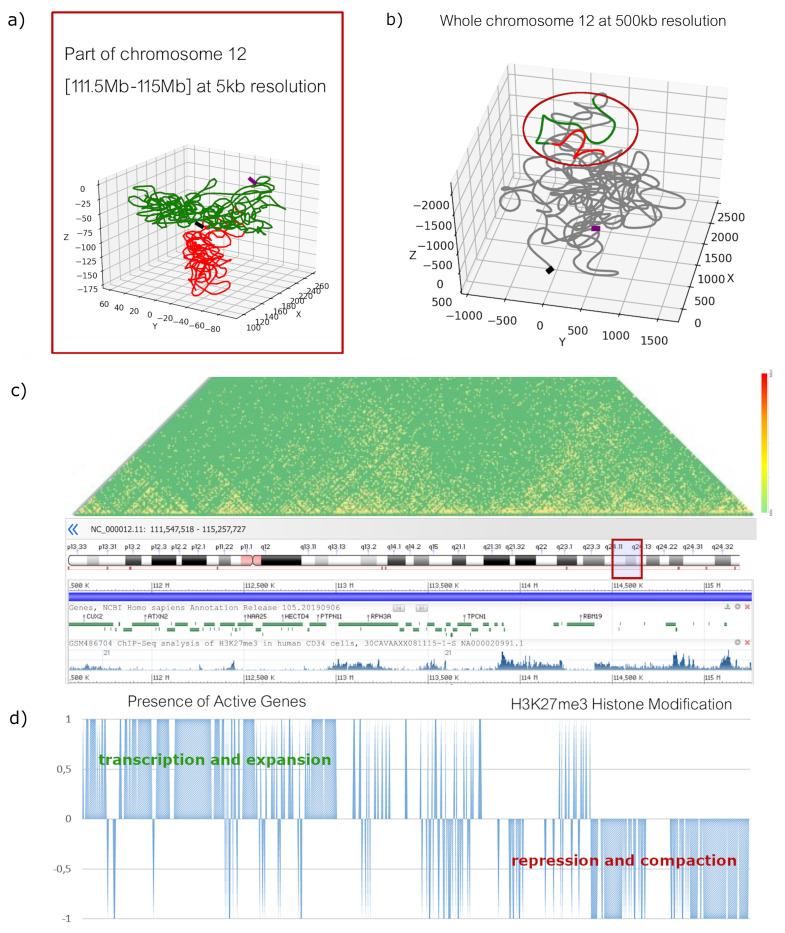
(**a**) Reconstruction of a portion of chromosome 12 [from 111.5 Mp to 115 Mp], at a 5 kb resolution (starting point in black, end point in purple); in green, the part of chromatin interested by active genes and more expanded; in red, the part interested by H3K27ME3, more compact. (**b**) Reconstruction of the whole chromosome 12 at a 500 kb resolution: the part in green, interested by active genes, is not only more expanded, but also outermost in the total chromosome. (**c**) Representation of HI-C, CHIP-seq and RNA-seq data referred to the same portion of chromosome 12 at a 5 kb resolution (plot from ENCODE). The areas with active genes show a lower concentration of H3K27ME3, while the areas with fewer genes, which are more methylated and more compact, correspond to higher HI-C contact frequencies (more yellow in the contact matrix heatmap). (**d**) Plot of CHIP-seq and RNA-seq information in *ChromStruct*’s input: 1, −1 or 0 score for every bin associated to expressed genes, H3K27ME3 and none, respectively (see Appendix A for details).

**Figure 4 biology-10-00338-f004:**
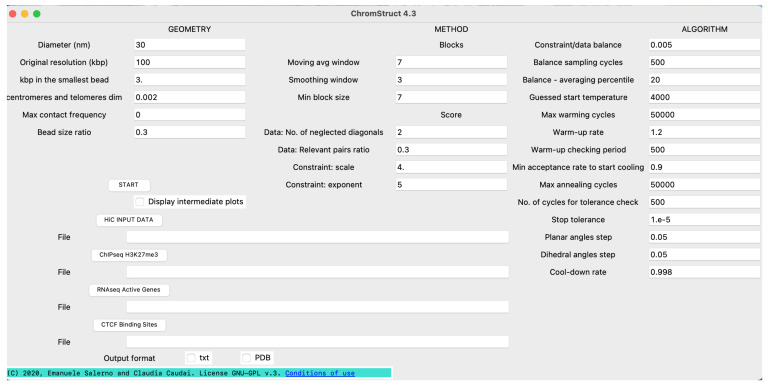
Graphical User Interface of *ChromStruct*. Three groups of quantities are displayed: the first (GEOMETRY) includes geometrical features, the second (METHOD) sets up the TADs extraction and the score function, and the third (ALGORITHM) is only related to the *Simulated Annealing* parameters.

**Table 1 biology-10-00338-t001:** Presence of structural information derived from H3K27ME3 CHIP-seq, RNA-seq and CTCF-binding experiments for 3.5 Mb portion of chromosome 12 [111.5 Mb–115 Mb], corresponding to blocks from 1750 to 1799 identified by block-detection algorithm.

Block	Dimension (kb)	Tot Contacts	Data	Corr 1 ${}^{a}$	Corr 2 ${}^{b}$
1750	75	272	Expr genes	0.128	0.134
1751	50	102	Expr genes	0.489	0.428
1752	65	295	Expr genes	0.246	0.191
1753	55	142	Expr genes	0.264	0.217
1754	100	133	Expr genes, CTCF	0.219	0.219
1755	70	41	Expr genes, CTCF	0.251	0.242
1756	65	158	Expr genes	0.358	0.295
1757	85	128	Expr genes, CTCF	0.286	0.217
1758	100	211	Expr genes	0.222	0.204
1759	45	89	Expr genes	0.320	0.371
1760	70	247	Expr genes	0.325	0.341
1761	70	153		0.113	0.185
1762	50	149		0.137	0.280
1763	55	178		0.322	0.364
1764	70	168		0.056	0.119
1765	100	228		0.133	0.161
1766	50	81	Expr genes	0.154	0.186
1767	45	163	Expr genes	0.346	0.255
1768	40	343		0.164	0.201
1769	55	268		0.222	0.160
1770	50	78	Expr genes, CTCF	0.326	0.204
1771	60	38	Expr genes	0.178	0.244
1772	110	389	Expr genes	0.303	0.235
1773	70	90		0.041	0.136
1774	100	637		0.230	0.178
1775	50	86		0.184	0.163
1776	80	383	H3K27M3	0.233	0.236
1777	45	77	H3K27M3	0.582	0.499
1778	60	143		0.306	0.318
1779	65	179	CTCF	0.408	0.342
1780	55	77		0.428	0.443
1781	85	66	CTCF	0.305	0.304
1782	105	249		0.324	0.241
1783	50	39		0.473	0.443
1784	85	218		0.330	0.313
1785	63	45	CTCF	0.209	0.210
1786	70	123		0.230	0.257
1787	45	104	H3K27M3	0.423	0.291
1788	70	283		0.145	0.131
1789	70	142		0.081	0.123
1790	45	80		0.202	0.224
1791	35	30		0.220	0.258
1792	65	185	H3K27M3	0.425	0.392
1793	70	208	H3K27M3	0.407	0.303
1794	50	53		−0.05	0.014
1795	40	168	H3K27M3	0.449	0.373
1796	140	1659	H3K27M3	0.250	0.286
1797	70	240	H3K27M3	0.266	0.155
1798	60	289	H3K27M3	0.233	0.150
1799	65	264		0.141	0.063

^*a*^ Pearson correlation between original Contact Matrix in input and synthetic Contact Matrix produced by
*ChromStruct* integrating HI-C, CHIP-seq, RNA-seq and CTCF data. ^*b*^ Pearson correlation between original Contact Matrix in input and synthetic Contact Matrix produced by *ChromStruct* using HI-C data only.

**Table 2 biology-10-00338-t002:** Pearson correlations between synthetic contact matrices and original HI-C contact matrix of the whole chromosome 12 for two populations of conformations: using HI-C data only (Experiment 1) and using HI-C, CHIP-seq, RNA-seq and CTCF-binding site data (Experiment 2).

	HI-C Contacts	RNA-seq	CHIP-seq	CTCF-Binding	Nr of Runs	Correlation ${}^{a}$
Experiment 1	✓				100	0.7188371
Experiment 2	✓	✓	✓	✓	100	0.6963284

^*a*^ Pearson correlation of the original HI-C contact matrix and the *ChromStruct*’s synthetic contact matrix at the first reconstruction-step resolution (average dimension of blocks is 800 kb).

## Data Availability

All the data used are publicly available from the URLs mentioned in the paper.

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
