# Peer review of "Integration of Multiple Resolution Data in 3D Chromatin Reconstruction Using ChromStruct"

_biology, 2021, doi:10.3390/biology10040338_

Round 1

Reviewer 1 Report

In this revised manuscript, Caudai and colleagues have addressed some of the concerns previously raised, the code can now be executed and there is some example data provided. However, several major issues remain, most critically the explanation and implementation of incorporating CTCF ChIP-seq data as a proxy for proximity in 3D space.

In their response, the authors state that the data they used is ‘experimentally-defined pairs of CTCF-bound sites through Immunoprecipitation with anti-CTCF antibodies. In other words: the proximity of the sites identified is confirmed by the sequencing of two fragments that were linked together in the experiments…’. This statement is confusing, and potentially incorrect.

It’s unclear what the authors mean by “experimentally-defined pairs” or “two fragments that were linked together in the experiments”. Are the authors referring to paired-end sequencing? The two ends of a paired-end fragment map at a distance from one another because the fragments sequenced are typically more than double the length of the sequencing reads. If CTCF is bound to any site along a long fragment, the entire fragment will be immunoprecipitated and the two ends of a paired-end read will not be immediately next to one another. However, this is not an indication of interaction between these two reads. Indeed, in the Methods, (page 8, lines 215-216), the authors state that ‘CTCF ChIP-seq contact features are introduced as a binary matrix’  but the output of a CTCF ChIP-Seq is not a matrix, and should be in the form of a 1D array (as stated for the H3K27me3 ChIP-Seq used in this study). How did the authors obtain this matrix? It seems like CTCF ChIP-Seq is treated as a conformation capture assay, but this would be improper.

Secondly, while it’s true that CTCF ChIP-seq can detect fragments that are bound by CTCF, the experiment does not tell anything about their physical proximity. CTCF has been involved in genome organization but its binding at a single site alone is not evidence of loop formation or proximity in 3D space with any other CTCF-occupied locus. Maybe I misunderstood the authors’ response, but this point should be clarified because using CTCF ChIP-Seq as evidence of proximity in 3D space is inappropriate.

I would still not recommend this manuscript for publication at this stage.

Author Response

Thanks to the reviewer for the comment and the advice. We have further clarified the origin of the data introduced in our study (link to the file from ENCODE project has been introduced in the text on lines 115 and 200). To address the concern of the reviewer, we further specify that in the case of CTCF, the data introduced are derived from ChIA-PET experiments. These experiments make use of antibodies to select DNA tracts that are bound to a specific protein (CTCF in our case) and that are ligated in situ. The Pair End fragments can map at a single locus, a frequent case due to self-ligation, or to loci that are distant in the genome. In the latter case, the result is attributed to the proximity, possibly induced by the bound protein.

In most cases these loci (identified by their sequence) are separated by a genomic distance of several tens of kb, thus making it very unlikely that the result can be attributed to the self-ligation of such a long fragment.

The introduction of this proximity has been modeled as a geometrical feature, interpreted as a possibility of closeness for two loci far from each other. These possible interactions have been modeled in the form of a square matrix (with binary values: 0 for bins with no interaction, 1 for bins with interaction), in a form that is equivalent to what is done for HiC contacts, because it encodes geometric features of the same type, namely the possible proximity of two loci in the three-dimensional space.

We would also like to stress that the main point of our work is the presentation of a method that enables the introduction of data from any source, as long as they provide information relative to chromatin geometrical features.

Reviewer 2 Report

The proposed changes have been addressed well. We would agree to publish this MS in present form.

Best regards

Author Response

.

Round 2

Reviewer 1 Report

The clarification that the authors have used CTCF ChIA-PET data to incorporate as proxy for physical proximity in their model addresses my previous concerns. However, I still disagree with the rationale behind using this data to add contacts to the Hi-C matrix. ChIA-PET is not an orthogonal approach to Hi-C for looking at contacts in 3D, as it is a method adapted from Hi-C. Thus, any contacts detected by ChIA-PET should be present in the Hi-C matrix anyway. Furthermore, detecting two fragments in proximity mediated by CTCF (as by CTCF ChIA-PET experiments) does not imply the potential interaction between these fragments is stronger or more stable than any other interactions detected by Hi-C.

I would still not recommend this manuscript for publication. However, if the editor wishes to accept it, it is critical that the authors state they used CTCF ChIA-PET data clearly throughout the manuscript and replace all references to ‘CTCF ChIP-seq’ or ‘CTCF binding site data’ (for example lines 88, 95, 103, 115, 117, 155, 216) as these techniques generate different types of data which also influences how their results are interpreted.

Author Response

The clarification that the authors have used CTCF ChIA-PET data to incorporate as a proxy for physical proximity in their model addresses my previous concerns. However, I still disagree with the rationale behind using this data to add contacts to the Hi-C matrix. ChIA-PET is not an orthogonal approach to Hi-C for looking at contacts in 3D, as it is a method adapted from Hi-C. Thus, any contacts detected by ChIA-PET should be present in the Hi-C matrix anyway. Furthermore, detecting two fragments in proximity mediated by CTCF (as by CTCF ChIA-PET experiments) does not imply the potential interaction between these fragments is stronger or more stable than any other interactions detected by Hi-C.

ChromStruct is designed with settable parameters so that each user can confer different weights to the geometric constraints derived from any experiment. It is, therefore, possible to consider the ChIA-PET interactions “stronger” than the Hi-C contacts, or equal or weaker, or not to consider them at all. This flexibility was designed to make the input and the balancing of information more versatile and to adapt to many different choices.

I would still not recommend this manuscript for publication. However, if the editor wishes to accept it, it is critical that the authors state they used CTCF ChIA-PET data clearly throughout the manuscript and replace all references to CTCF ChIP-seq or CTCF binding site data(for example lines 88, 95, 103, 115, 117, 155, 216) as these techniques generate different types of data which also influences how their results are interpreted.

Thanks for the advice, we have changed the nomenclatures CTCF ChIP-seqandCTCF binding site data with ‘CTCF ChIA-PET data’.

This manuscript is a resubmission of an earlier submission. The following is a list of the peer review reports and author responses from that submission.

Round 1

Reviewer 1 Report

Summary

The authors present here an extension of a previously published tool to infer chromatin 3D structure. This extension allows the integration, as modelling features, of chromatin related features like ChIP-seq or RNA-seq, and, as their previous version, the modeling is meant to be multiscale.

Major comments

General comments

I have a major concern regarding the results presented in this paper. The author state that including data about histone marks, expression and CTCF they are able to improve their models, but I don’t see it to be true. The authors acknowledge this fact at low resolution (Table 2), but they somehow don’t show it at high resolution. The authors only show comparisons of their modeling when using data separately, but they skip showing the result of their modeling when taking into account all data together. And it seems that taking all data together does not improve the 3D modeling of the chromatin in the only example they show at 5kb resolution (see further comments below). I strongly encourage the authors to review and complete this part of the manuscript, or to prove me wrong. 

The authors include in this version of their software data about the epigenetic state of the chromatin  compactness. They explain how they expand or shrink beads by 50% in euchromatin. But it is not clear to me how they are able to control this size using only the curvature parameter and I don’t see any parameter related to bead diameter in the formulas (it seems to be present in the code though). I would also like to acknowledge that they compare or discuss previous studies that have shown the relationship between chromatin 3D structure (dihedral angles, densities...) and epigenetic states (Serra et al. 2017) (please, if you want to cite one of such work use other references, I could not remember alternatives now).

The authors compare their synthetic contact matrices to the input Hi-C data and, at high resolution, show correlations around 0.2 or 0.3. I am not sure how this correlation is made (see comment bellow), but this seems very low, I wonder if this value is better than doing the 3D modeling with no Hi-C data input (only restraint being chain connectivity, and then brownian motion). 

Specific comments

Figure 2, and related text: I understand that synthetic contact matrix refers to the contact matrix produced by the 3D models of the chromatin, but it is not stated clearly in the text. I am not sure about the last box plot comparison “HiC contact frequencies”, does it simply represent different seeds? If so, perhaps some statistical test of the differences could be displayed in each pair of boxplots to reinforce the message. Also it is not clear to me why this work is done only at this specific region of chromosome 12. What happens if the authors attempt to do the same on the whole chromosome 12?

Also regarding Figure 2, what correlation is used? Is it a Pearson correlation between the contact matrices? Is it averaged between the 100 models?

Finally, there is one boxplot comparison missing, the one shown in Table 1, with all experiments vs only Hi-C. I quickly did this comparison using the values in the table (corr 1 vs corr 2) and found no significant differences (Mann-Whitney p-value: 0.2). The authors should show and address this issue. This comparison is done at 500kb resolution in Table 2, I would expect the same for this finer resolution.

Figure 4: I don’t understand the figure, either the labelling of the third box-plot “HiC-only”is wrong, or these are contacts from synthetic matrices… 

In the source code: potential typo line 293, “if (axisqua[0]==0. and axisqua[0]==0. and axisqua[0]==0.”) this checks nullity 3 times on the same variable. I would recommend the authors to provide a test script in order to check if everything is functioning as expected.

Minor comments

I am missing information about the HiC data used, cell type, density (e.g. number of valid pairs). Something a bit more specific than “human hematopoietic cells at time T0”.

Typo p2 line 58 “form”

Figure 3a,b: what are the black/purple rectangles? I guess begin/end?

p7 , line 140: what result is to be expected?

P8: 189: where are the R scripts?

francois serra

Reviewer 2 Report

In this manuscript, Caudai and colleagues present an improvement of their 3D genome modelling software, ChromStruct. The latest implementation of their modelling approach has been previously described in detail in Caudai et al. (IEEE, 2018). There, the authors explain the rationale of their method, which aims to model 3D chromatin structure based on Hi-C interaction frequencies.

Here, in addition to using Hi-C interaction data, the authors take advantage of H3K27me3 ChIP-seq, RNA-seq and CTCF binding site data to introduce additional modelling constraints. The authors take H3K27me3 ChIP-seq and RNA-seq as proxies for more or less tightly packed chromatin, respectively, and incorporate this information in their model. In order to validate their approach, the authors evaluated the correlation between the Hi-C data that was used for fitting and the one inferred from the model. When introducing additional constraints based on ChIP-Seq, RNA-Seq or CTCF binding data, the correlation shows a small improvement at 5 kb resolution (but remains low overall). At larger scales, incorporating this information does not seem to improve the model performance (Table 2). Instead, the correlation slightly decreases. Given these results, it’s unclear whether this implementation can be really considered as an improvement over the previous version.

Perhaps more critically, ChromStruct predictions have never been tested against other methods, benchmarked with simulated data or tested in different biological settings. The method was only fitted for one sample (for which the Hi-C data is not publicly accessible, and RNA-Seq and ChIP-Seq data are not referenced). The underlying model may be of interest from a theoretical perspective, but without further tests it’s impossible to judge whether it can be used to draw meaningful biological conclusions. Examples include e.g. comparing the results between replicates (to prove that predictions are robust) and and in conditions where chromatin structure is known to be altered (e.g. depletion of cohesin complex subunits like Rad21, Nipbl, Wapl, to show that it can capture changes in chromatin structure). Finally, there are some issues with the execution of the code, and potential concerns with the way CTCF information is incorporated.

In conclusion, I would not recommend the manuscript for publication at this stage. A more detailed list of recommendations is provided below.

Major points:

  • The authors should compare the consistency of models generated from replicate Hi-C experiments to demonstrate the robustness of their approach. They should also compare models of regions that change between biological conditions to showcase these are reflected in their predicted structures. ChromStruct results should also be compared to different methods for modelling 3D chromatin structure.
    The authors state that ‘CTCF ChIP-seq data inform about two DNA segment that are in close contact with some frequency’ (page 9, lines 195-196). This statement is incorrect, as CTCF binding at certain regions does not imply their physical proximity. While CTCF is commonly found at TAD boundaries, this is the case only at 70% boundaries (Nora et al., 2017), and furthermore, only ~10-15% CTCF binding sites correspond to TAD boundaries/loop anchors (Dixon et al., Nature, 2012, Fig. 2C, Bonev et al., Cell, 2017, Suppl. Fig. S3C), mainly when in convergent orientation. More critically, the Supplementary README file suggests that the presence of CTCF at a given site was used to artificially change the counts in the Hi-C contact matrix (cf. “CTCF contacts need to be introduced in the contact matrix as additional contacts with high frequency (set equal to the maximum frequency of the matrix by removing the main diagonal)”). If this is the case, this would be totally inappropriate, since the presence of CTCF is not a direct evidence of contact. The authors should clarify this further and better justify their rationale.
  • The origin of the RNA-seq, H3K27me3 ChIP-seq and CTCF binding site data should be clearly specified. The EGA entry (EGAS00001001911) that the authors refer to (page 8, lines 184-185) only describes Hi-C data and it’s unclear what is the source of the RNA-seq or ChIP-Seq data (also it’s hard to know because the data set is not publicly accessible). Furthermore, the authors later refer to CTCF ChIP-seq (page 9, lines 195-196), which they have not previously mentioned.
  • In the Results (page 3, lines 98-99), the authors state they took ‘RNA-seq, H3K27me3 ChIP-seq data at 20bp resolution’, but in the Methods (page 9, lines 193-195) they explain the ChIP-seq was binned at 5kb to correspond to the Hi-C resolution, whereas the RNA-seq was mapped to genes. The authors should clarify how the RNA-seq, ChIP-seq and Hi-C data were processed and normalized (unclear what ‘Bonferroni correction’ (page 8, line 189) refers to in this context) and what cut-off was used to consider genes as expressed.
  • The authors should also clearly state the Python dependencies required for their script to run, as well as the exact format of the input Hi-C matrix. Ideally, example data should be provided for trial runs of the script. Indeed, I could not get the code to run fully. After running the script with a Hi-C matrix (ICE normalized, dense matrix in .txt format without row and column names) of human chromosome 18 at 150kb, 40kb or 10kb resolution, either with or without a ‘simulated’ ChIP-seq track per bin, the script only detects one block size and only generates the following files (…_0_BlockSizes.txt, …_Log.txt, …_centromeres.txt, …_telomeres.txt). No file with a table that demonstrates if a sub-chain intersects H3K27me3/RNA-seq signal is described in the Supplementary or was generated by the code, as is referred in the main text (page 4, lines 111-112). After this, the code produces the following error in Python v3.7.3 with matplotlib v3.3.3 (and no plots were generated):
    Traceback (most recent call last):
    File "/.../.../bin/ChromStruct-4.3/ChromStruct_4.3.py", line 1719, in <module> chromstruct(hm,conf,bigblocks) # Call iteration
    File "/.../.../bin/ChromStruct-4.3/ChromStruct_4.3.py", line 1500, in chromstruct output(C)
    File "/.../.../bin/ChromStruct-4.3/ChromStruct_4.3.py", line 1405, in output ax.set_aspect('equal')
    File "/.../.../anaconda3/lib/python3.7/site-packages/mpl_toolkits/mplot3d/axes3d.py", line 324, in set_aspect "Axes3D currently only supports the aspect argument "
    NotImplementedError: Axes3D currently only supports the aspect argument 'auto'. You passed in 'equal'.

Minor points:

  • In Figure 2, the authors should explain what the HiC Contact Frequencies (bottom right boxplots) set is and how the two sets of ‘HiC only’ differ. They should also clarify if the values used in the boxplots are from the different segments of the single simulation described in Table 1, or from several models.
  • The authors should explain what the correlation coefficients between the individual synthetic contact matrices and the original matrix are and how they were calculated (Figure 2, Table 1).
  • The block-detection algorithm described in Caudai et al. (IEEE, 2018) that the authors refer to in the Results (page 3, line 108) should be summarized here to ensure this manuscript can be understood on its own.
  • Throughout the manuscript, the authors refer to blocks being ‘interested’ by H3K27me3/RNA-seq (page 4, lines 113, 116, 122, page 7, line 144, page 9, line 198, page 10, Figures 2, 3, 4). Do they mean ‘intersected’?
  • When referring to TADs, the authors state they are ‘sometimes described as fairly isolated ‘globular’ structures’ (page 1, lines 32), which is a somewhat outdated view (recently reviewed in Beagan and Phillips-Cremins et al., Nat Gen, 2020)
  • The authors state that ‘In addition to Hi-C, … ChIP-seq … and RNA-seq can provide information on geometrical features of chromatin’ (page 2, lines 34-35). However, while gene expression and histone PTMs can be correlated with the extent of chromatin compaction, it is yet unclear how they are associated with 3D genome conformation.
  • The second sentence of following statement is ambiguous and should be more clearly phrased: ‘… expressed genes are frequently marked by … H3K27ac … in their enhancer … and H3K4me3 in their promoter region, while repressed genes are often marked by K3K27me3 in their body. However, not all expressed genes bear such mark, and not all marked genes are necessarily expressed.’
  • The authors may wish to acknowledge that while H3K27me3 is a proxy for facultative heterochromatin, it will not allow taking into account constitutive heterochromatin, which is marked by H3K9me3. To get more information about heterochromatic regions, they could use LMNB1-DamID data that reveals chromatin proximity to the nuclear lamina, which is associated with repression.
  • The authors should add a scalebar to the heatmap in Figure 3C.
  • In Figure 4, it appears that the authors are plotting contact density per block calculated from the original Hi-C data they used for their modelling. Thus, what is represented in the figure does not concern their model, but simply reflects a property of the chromatin organization of the region the authors are focusing on and does not contribute to the results described in the rest of the manuscript.

Reviewer 3 Report

[Summary]

The authors describe an extension of their software tool for
construction of 3D chromatin conformations from Hi-C data, named
"ChromStruct". The extension allows to include data from ChiP-Seq and
RNA-Seq experiments. It is shown that this inclusion improves the
results for high resolution reconstructions while the quality for lower
resolution stays the same.

[Opinion]

Including data from experimental sources like ChIP-Seq and RNA-Seq into
chromatin reconstructions seems reasonable and the described algorithm
seems effective for that. To my current knowledge, comparable tools,
using data from several other experiments for the 3D reconstruction of
chromatin, do not exist.

Accordingly, I think that the described extension of ChromStruct could
be a powerful tool for investigating chromatin structures with higher
resolution than currently possible.

Before publication, I see a moderate number of minor corrections of the
text necessary. Some of the statements are too ambiguous and should be
corrected to be more precise and clear (See below for details).

[Extensions]

I recommend giving a short review on the impact of histone modifications
and transcription on chromatin compaction as done in 4.2. Maybe move
some of the lines from there to the introduction. These findings are
crucial for the motivation of the whole algorithm and should therefore
be part of the introduction.

I would also recommend adding one or two lines on the input and results
of "ChromStruct" to the introduction. First, if a fixed consensus
structure is modelled or a more dynamic picture. This is crucial, since
in [32-33] the dynamic nature of TADs was mentioned while it is not
discussed if these dynamics were considered. Second, if the tool
requires single-cell Hi-C data or if bulk data is sufficient.

[Corrections]

[26-27] "how frequent" would be more technical than "how often"

[31-32] "many" and "few" is subjective. I would recommend giving a
relation like: "more internal ... compared to external interactions".

[36, 40] "tracks" seems unconventional in this context and more related
to filetypes than parts of chromatin. I would recommend a term like
"region", "locus" or "part".

[43-44] It is plausible that the data is redundant but either a source
is needed for the statement or it should be reformulated as (plausible)
assumption or expectation.

[51] Is "also" necessary?

[93] The "Bayesian" approach should be explained or it should be
referred to the original ChromStruct paper. The term "cost function"
should be exchanged by "score function" for consistency with the later
use of the term in Methods section.

[100] Why exactly 100 conformations?

[118] Since it is a crucial part of the algorithm, a short description
of the inclusion of CTCF data should be part of the main paper.

[131-132] "too fine" seems ambiguous to me. Should be described or moved
to discussion.

[140] ... was expected (?)

[140] "agreement": I think "overlap" or "redundancy" was meant.

[159] "... by HI-C contact matrices (+alone)."

[171-173] more explanation on "reinforcement of Bayesian approach"
needed or source for the statement.

[178] "consistency": I think "overlap" or "redundancy" was meant.

[186] "transform" seems strange in the context of an alignment.

[189] source for the use of "Bonferroni correction" on Hi-C data needed
or explanation

[???] "acts on the curvature of a sub-chain". "curvature" seems strange
here. Reformulation or explanation?

[???] I would like to see a motivation for the particular form of the
terms for ChIP and RNA-Seq inclusion. Were other possibilities tested?

[???] I would like to see more motivation on why beads are not allowed
to penetrate each other by more than a certain fraction.

[226-235] Is a summary at this point necessary?

[236] I am missing a clear statement like "the results show that the
precision of chromatin reconstructions can/cannot be significantly
improved by the ChromStruct extension."